# Compound Extremes of Droughts and Pluvials: A Review and Exploration of Spatio-Temporal Characteristics and Associated Risks in the Canadian Prairies

Elaine Wheaton [1,*], Barrie Bonsal [2] and David Sauchyn [3]

1    Department of Geography and Planning, University of Saskatchewan, Saskatoon, SK S7K 1M2, Canada
2    Watershed Hydrology and Ecology Research Division, Water Science and Technology Directorate, Environment and Climate Change Canada, Saskatoon, SK S7N 3H5, Canada; barrie.bonsal@ec.gc.ca
3    Prairie Adaptation Research Collaborative, University of Regina, Regina, SK S4S 0A2, Canada; david.sauchyn@uregina.ca
*    Correspondence: elainewheaton@sasktel.net

**Abstract:** The Canadian Prairies are associated with high natural hydroclimatic variability including the frequent periodic occurrence of droughts and pluvials. These extremes carry various risks including significant damage to the economy, environment and society. The well-documented level of damage necessitates further risk assessment and planned reductions to vulnerability, particularly in light of a warming climate. A logical starting point involves awareness and information about the changing characteristics of such climate extremes. We focus on the compound occurrence of droughts and pluvials as the risks from this type of event are magnified compared to the hydroclimatic extremes in isolation. Compound droughts and pluvials (CDP) are drought and pluvial events that occur in close succession in time or in close proximity in area. Also, research on CDP is limited even for the worldwide literature. Therefore, the purposes of this paper are to synthesize recent literature concerning the risks of CDP, and to provide examples of past occurrences, with a focus on the Canadian Prairies. Since literature from the Prairies is limited, global work is also reviewed. That literature indicates increasing concern and interest in CDP. Relationships between drought and pluvials are also characterized using the SPEI Global Monitor for the Prairies, emphasizing the recent past. Research mostly considers drought and pluvials as separate events in the Prairies, but is integrated here to characterize the relationships of these extremes. The spatiotemporal patterns showed that several of the extreme to record pluvials were found to be closely associated with extreme droughts in the Prairies. The intensities of the extremes and their dry to wet boundaries were described. This is the first research to explore the concept of and to provide examples of CDP for the Prairies and for Canada. Examples of CDP provide insights into the regional hydroclimatic variability. Furthermore, most literature on future projections strongly suggests that this variability is likely to increase, mainly driven by anthropogenic climate change. Therefore, improved methods to characterize and to quantify CDP are required. These findings suggest means of decreasing vulnerability and associated damages. Although the study area is the Canadian Prairies, the work is relevant to other regions that are becoming more vulnerable to increasing risks of and vulnerabilities to such compound extremes.

**Keywords:** droughts; pluvials; excessive moisture; extreme climate events; compound drought and pluvials; compound extremes; climate change; risk; vulnerability; Canadian Prairies



## 1. Introduction and Objectives

Canada is faced with an increasing risk of drought and excess precipitation in a warming climate (e.g., [1–3]). Within Canada, the Prairie region stands out as a hotspot for both droughts and pluvials and therefore is associated with significant risks from these disasters. For example, a recent natural hazard assessment for Saskatchewan found

that droughts and convective summer storms had the highest aggregate risk levels [4]. The Prairies have more exposure to these extremes because of the high variability of precipitation in both time and space [5]. Often, periods and areas of drought and excess moisture occur in close proximity. Since hydroclimatic extremes are projected to occur with greater frequencies and intensities in a warming climate, more compound occurrences can be expected.

The selected study area for this paper includes Canada's major agricultural region within the provinces of Alberta, Saskatchewan and Manitoba, termed Prairies here (Figure 1). In this region, droughts cause considerable damage to ecosystems, the economy, human health, and society. Drought can be defined as a prolonged period of abnormally dry weather that depletes water resources for human and environmental needs [6]. More specific definitions depend on the measure used, as well as the application, users, and regions. Droughts are classified into several different types, including meteorological, hydrological, agricultural, ecological, and socioeconomic [7]. Borowski adds another type, hydrogeological drought, which refers to a long-term decline in groundwater resources [8]. We consider some of the more severe cases of drought in the Prairies, as indicated by specific water balance values.

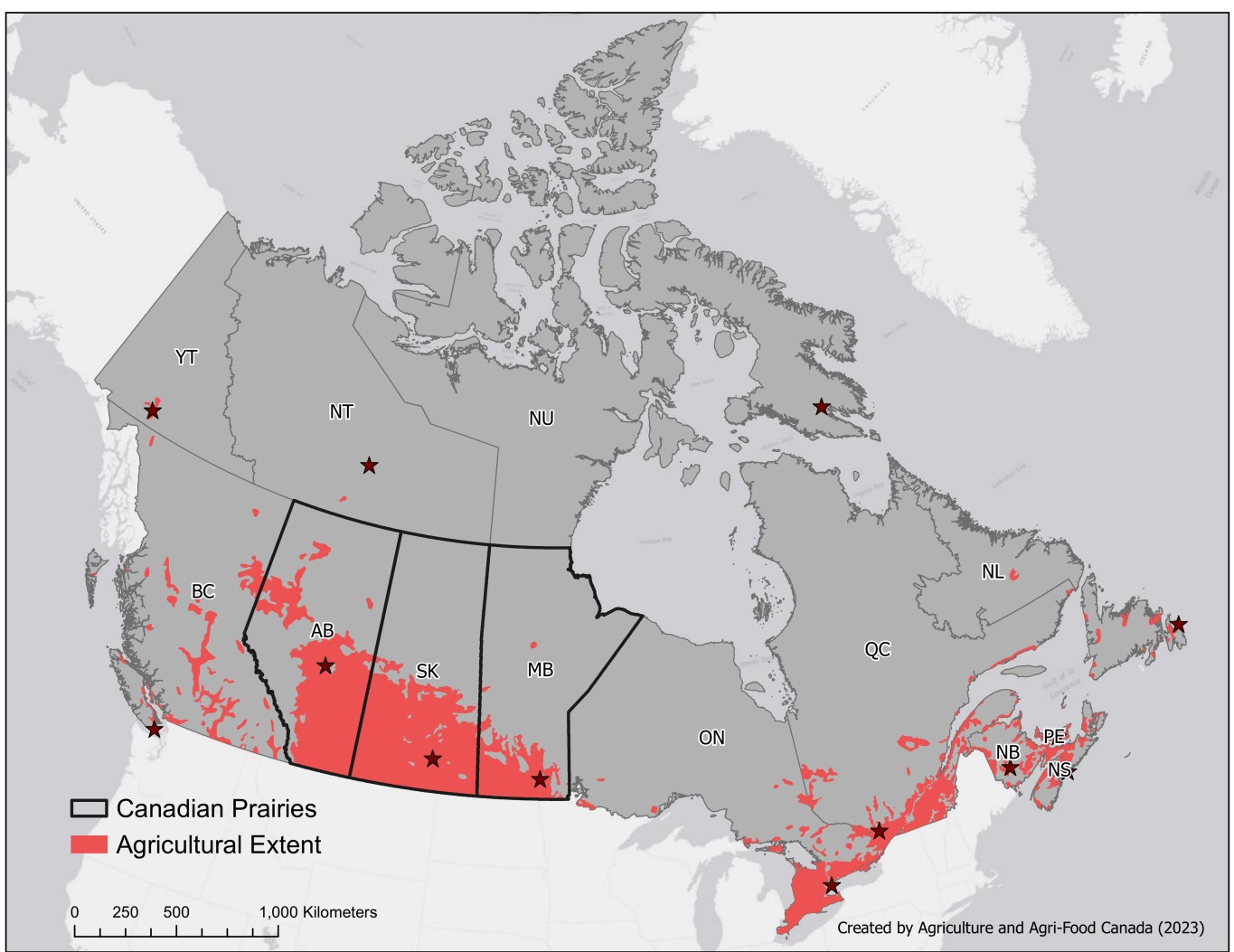

**Figure 1.** Study area map, the agricultural region of the Canadian Prairie Provinces of Alberta (AB), Saskatchewan (SK) and Manitoba (MB) (Reprinted with permission from Agriculture and Agri-Food Canada, 2023). The stars show the location of the capital cities of the provinces.

A well-documented example of the damage from droughts is the 1999–2005 drought which is considered one of Canada's worst natural disasters. It caused the Canadian economy to suffer a loss of $4.5 billion in 2001 to 2002 alone, with most of the damage concentrated in the Prairies and in agricultural production [9]. More recent droughts have also caused much damage in the Prairies, including the drought and heat dome of 2021, which encompassed about 99% of the agricultural prairies and resulted in considerable damage and hardship [10]. Another example of damage from both droughts and pluvials is their role in the declining environmental sustainability of agriculture [11].

Pluvials, that is intense and/or extreme rainfalls, also cause damage and loss. We expand the term here to include extreme precipitation and wet spells of various sorts similar those described in He and Sheffield (2020) [12]. Historically, pluvials have caused less damage than droughts, likely because droughts often extend over larger areas and longer times, and therefore, affect more people, assets and places. The extensive areal coverage of droughts has been documented by Wheaton et al. [9], for example, for Canada and even large parts of North America. Despite their lesser extent, pluvials and their associated floods still have resulted in considerable impacts, including threats to safety, and damage to infrastructure, reservoirs and agriculture. Wittrock et al. documented many costly impacts in prairie communities including flooding of homes and businesses, compromised drinking water, damaged roads and overwhelmed emergency services [13]. Flooding hazards in the three Prairie Provinces resulted in the largest payouts from the Federal Disaster Financial Assistance Arrangements during 1970 to 2014 [14]. The estimated cost of flooding in June 2010 was over $1B. The most severely affected areas were southern Alberta and Saskatchewan where over 2000 people were evacuated. Heavy rains also caused serious flooding in 2011 in southeastern Saskatchewan and southern Manitoba [15].

Risk consists of the combination of the probability of occurrence of a hazard and the consequences of a hazard [16]. The IPCC (2012) defines disaster risk as the "likelihood... of severe alterations in the normal functioning of a community or society due to hazardous physical events interacting with vulnerable social conditions leading to widespread adverse ...effects..." [17]. Both the likelihood and adverse effects of droughts and pluvials are high for the Prairies, resulting in elevated levels of risk. A starting point for adaptation preparedness and vulnerability reduction is awareness and information about the changing characteristics of such climate extremes.

Research regarding the compound and complex interactions between droughts and pluvials appears as a gap in the world-wide literature (e.g., [12,18,19]). Usually, these extremes are assessed individually and switching of extremes in time or area is not usually considered. The IPCC (2012) defines compound events as two or more extreme events occurring simultaneously, in close succession, or concurrently in different regions [17]. They describe various compound events, including drought and heat, heat and precipitation extremes, drought and fire, and precipitation and wind, for example, but they do not seem to consider the convergence of drought and precipitation extremes in time or area. The purpose of this paper is to address this gap by means of literature reviews and exploration of past compound drought and pluvial events in the Prairies.

The IPCC (2021) stated that the probability of compound events has likely increased in the past due to human-induced climate change and will likely continue to increase with further global warming, and that multiple compound extremes can lead to stronger impacts than those experienced in isolation [20]. However, a discussion of the compound occurrence of drought and heavy rains or excess moisture is missing in that source and in most other literature. An assessment of these compound events is critical as they can result in multiple stressors that quickly exceed the coping capacity of systems. Several authors stressed that the consecutive occurrence of dry and wet extremes can have larger social and environmental impacts than isolated events (e.g., [12,18,19]). Rezvani et al. noted that these compound extremes may be more challenging for water managers [21]. However, if pluvials alleviate droughts by replenishing reservoirs, wetlands, soil moisture and groundwater, the interaction can have positive impacts. Therefore, droughts and

pluvials may have partial offsetting impacts depending on the amount of rainfall, event size, order, location and timing of the events. With their compound nature, however, these extremes can be a threat, impactful and even vulnerability multipliers and magnifiers.

The main research questions we address are: What literature has examined the compound characteristics of both droughts and pluvials? What are some examples of compound droughts and pluvials (CDP) in the Prairies? Our objectives are to (1) synthesize recent literature concerning the past and future possible risks of interacting droughts and pluvials, with a focus on those papers relevant to the Canadian Prairies, and (2) to provide and describe examples of Prairie CDP. We do this in two ways, one, by integrating the literature that examines droughts and pluvials separately, and two, with examples identified using the Global SPEI (Standardized Precipitation and Evapotranspiration Index) Monitor for the Prairies. The overall purpose is to improve the understanding of CDPs with special reference to the Prairies.

## 2. Methodology

The methods used were to (1) review the literature using synthesis and inter-comparisons, (2) use the findings from Prairie studies that consider droughts and pluvials (DP) independently and combine them to document the combined occurrence of the extremes, (3) use the Global SPEI Monitor [22] to describe examples of compound DP in the Prairies. These approaches also increase the understanding of the nature of hydroclimatic variability in the recent past. We focus on the 2000 to 2022 period for ease of access to data and for relevance. The second and third approaches use a combination of qualitative, quantitative and case study methods (e.g., [8,23]). All these methods were used, as appropriate, for this exploratory stage of CDP assessment for the Prairies.

Relevant literature is described and findings are synthesized to assess implications for risk from CDP. Literature regarding both past conditions and future projections is considered given the expectation that they increase with continued climate change, and how adaptation for pluvials will improve adaptation to drought, such as water storage, improved infiltration, and less drainage are also studied in that literature. Perception and experience associated with these extremes play a role in adaptation and vulnerability. For example, Marchildon et al. found that wet periods tend to reduce the memory experience and awareness of drought risks exacerbating human vulnerability [24]. This finding is yet another reason that these extremes should be considered together for reducing vulnerability.

Because of the lack of CDP literature for the Prairies, we broadened the geographic scope of the search. We emphasize literature for North America and globally, but for relevance to the Prairie climate regions. The assessments of CDP in the Prairies are few [25–28]. Therefore, we use a convergence of the more common separate studies on DP to explore examples of CDP in the Prairies.

Pluvials are often indicators of the nature of the stages of droughts. They can determine how quickly droughts start and grow, and end, the type of persistence stages or even where they migrate. In terms of spatial patterns, pluvials may also be found adjacent to drought areas. We used literature on pluvial extremes in the Prairies to begin this integration process of exploring CDP. Five pluvial examples were selected to explore the nature of the relationship with possible drought occurrence at that concurrent time and place. The spatiotemporal relationships of the extremes are then described.

Both DP extremes can be measured by various quantitative water balance indicators as these indicators are considered necessary for drought monitoring and assessment (e.g., [29]). These indices include the commonly used Standardized Precipitation Evapotranspiration Index (SPEI), Standardized Precipitation Index (SPI), and the Palmer Drought Severity Index (PDSI). Other indices are being developed, such as the Standardized Palmer Drought Index [29].

To characterize examples of the compound nature of DP events in the Prairies for selected events in 2000–2022, we use the Global SPEI Monitor [30]. The target period is September to August, the agricultural year, and the associated 12-month SPEI is the August

value. The previous method began with the dates of extreme pluvials from the Prairie literature. For this part, we begin with dates of known droughts from the literature. Eight examples were selected that were years of severe drought. The patterns of the SPEI were examined and the main areas of drought and of wet conditions were described, along with their intensities. Then the relationships of the CDP events were described, including the gradients between them and the orientations of the wet to dry boundaries.

A longer period has the advantage of capturing the more severe events. This approach supplements the method of the convergence of findings from separate assessments of DP in the literature for the Prairies. The characteristics of CDP should also be explored on a shorter time scale as use of the annual scale can smooth over shorter-term important fluctuations, especially for pluvial events. This is a limitation of using this time scale. As drought and excessive moisture events are known to have decadal variations (e.g., [31]), even longer time scale patterns of CDP are also worth assessing.

A limitation of the use of the Global SPEI Monitor is that the Thornthwaite method is used to calculate the potential evapotranspiration (PET) component of the SPEI. This method has a temperature bias, and thus is considered less accurate than more complex methods. It is often used for simplicity of calculation and enables the advantages of accessibility and near-real time character. Real-time data sources for more robust PET estimations are lacking and also require larger datasets. The Monitor has several advantages, however, including availability of several time scales for the SPEI, of specific grid values, for considering both wet and dry times, interactive ability, one-degree spatial resolution, time series ability, several time scales, and access to global areas, to name a few. Temperature is also included in the calculations of SPEI, rather than just precipitation, and this is necessary in addressing the increasing role of climate change.

The computation of SPEI and other information are described at the website for the Global Monitor (https://spei.csic.es/map, accessed various dates July to September 2023). Because the index is standardized, the average value of the SPEI is zero and the standard deviation is one. The standardization makes it comparable over time and space. Another advantage of the Monitor is that values of SPEI for each grid are accessible by using the "mouse-over" feature. Examples of the ranges of values of SPEI are shown in the time series of water-year SPEI for locations on the Prairies. These ranges are in the order of values as low as about $-2.5$ to near 2.5 [32].

Therefore, this Monitor is suitable for our purposes of examining the main spatio-temporal patterns of selected CDP. In contrast, other monitors such as the North American and Canadian Drought Monitors are available only for the monthly scale and do not provide grid values, for example. Most importantly for the purposes of this paper, they do not include information for pluvials. Therefore, they are less suitable for exploring CDP characteristics.

## 3. Results

### 3.1. Literature Reviews and Synthesis

Decadal to multi-year hydro-climatic variability tends to dominate the time series of drought and excessive moisture in the Prairies [31,33]. This variability occurs over a background of significant warming especially in winter and spring [3]. Bonsal et al. documented increases in inter-annual variability using the 30-year standard deviation of summer SPEI across the southern Canadian Prairies [32]. That variability is an indicator of dry-wet switching.

Observations and simulations of the hydroclimate of the 20th and 21st centuries have a record-length limitation for capturing low-frequency variability. Proxy climate records of the past millennium, on the other hand, reveal the persistence of this scale of variability and its dominance prior to the period of greenhouse gas warming [34,35]. For example, Kerr et al. used a network of more than 80 tree-ring chronologies to develop independent reconstructions of the warm and cold season flow of the North and South Saskatchewan Rivers since 1400 [36]. Thus, they were able to document how often drought and pluvials

occur over a very large part of the prairies and conversely when excess water in one basin offsets water deficits in the other one. They found that there are more similarities between basins during wet periods than dry. This points to the patchy distribution of dry conditions and the importance of local convective precipitation as a water source in dry years. Pluvials, on the other hand, generally correspond to the availability of water from large-scale mid-latitude cyclones [37].

Although most research assesses DP events separately, these studies examined some forms of CDP in the Prairies [25–27]. That research was carried out for the Drought Research Initiative and for the 1999 to 2004 drought [38]. These three papers appear to be the main research examining CDP in the Prairies, to the authors' knowledge. Shabbar et al. identified extreme wet and dry seasons of the growing season (May to August) from 1950 to 2007 using the Palmer Z-Index [26]. Their main focus was the analysis of the interrelationships among large to synoptic-scale atmospheric circulation patterns and cyclone characteristics during extreme drought and pluvial periods in the Prairies. Their Z-index time series shows considerable interannual and some multi-year variability of the time series with much wet-dry switching.

Evans et al. documented precipitation during the 1999–2004 drought at sites in Alberta and Saskatchewan [27]. They found that precipitation events of daily accumulations of 10 mm or less accounted for up to 63% of the total precipitation at these sites during that drought. They state that any understanding of drought must consider precipitation issues.

Szeto et al. characterized the catastrophic June 2002 Prairie rainstorm that occurred during the major drought of 1999 to 2004 [25]. Many locations experienced record-breaking rainfall amounts and major flooding. They showed that atmospheric conditions associated with the extreme background drought enhanced the likelihood of the co-occurrence of DP and facilitated the development of the extreme pluvial. This tremendous pluvial alleviated the drought conditions in the southern Prairies and contrasted with the still severe drought in other areas.

Brimelow et al. characterized the exceptional variability of precipitation patterns in the Canadian Prairie Provinces from 2009 to 2011 [28]. They found rapid transitions of drought to pluvial in both time and space. They show three main areas of drought and pluvial patterns in our study area during that period. Their study area extended farther north than ours, and they describe the contrast between very dry conditions in the boreal zone compared with wet conditions over far southeastern Saskatchewan and southern Manitoba.

A second type of literature of global scope demonstrates existing work on compound extremes and more specifics about the spatial and temporal patterns and methods used (Table 1). Although the review centres on North America for relevance to the Prairies, we expanded it to include the global scale, given the scarcity of literature for our study area. The literature found for North America was for the Northern Great Plains and Midwest areas of the United States [39,40] and for the southeastern US [41]. Rezvani et al. assessed the Fraser, Columbia and Peace River Basins in British Columbia and a portion of northern Alberta [21].

**Table 1.** Overview of selected literature for the comparison and synthesis of findings for the compound nature of drought and pluvial events. References are ordered by year.

| Reference | Compound Extreme Type | Description /Finding | Temporal and Spatial Patterns | Region | Methods |
|---|---|---|---|---|---|
| Christian 2015 [39] | Drought year followed by a pluvial year (dipole) | Chance of significant pluvial year after a significant drought year about 25% | Autumn to early winter period is critical to transitions Doubling of dry–wet events in more recent observation period 1955–2013 | Northern Great Plains of US | Standard deviations of drought to pluvial transition to define dipoles, hydrological year |

**Table 1.** *Cont.*

| Reference | Compound Extreme Type | Description /Finding | Temporal and Spatial Patterns | Region | Methods |
|---|---|---|---|---|---|
| Martin 2018 [42] | Projections of pluvial and drought characteristics, number, duration, severity | Worsening droughts and pluvials most apparent in N Hemisphere mid latitudes and Americas | Drying regions may see more longer and stronger pluvials Precipitation variability expected to increase globally | Global | SPI 6month, CMIP5 models, severity index using SPEI thresholds and time |
| Maxwell et al., 2017 [41] | Pluvials resulting in drought termination | 73% of droughts ended rapidly in a one-month period | Rapid drought endings are more common than gradual endings | Southeastern US | PDSI, moderate level, percentage of grids, storm classification, trend analyses, oceanic and atmospheric indices |
| He Sheffield 2020 [12] | Lagged compound droughts and pluvials | 11% of droughts followed by pluvials, with rapid transitions | Drought to pluvial transitions have increased in past 30 years | Global, with Western Canada noted to have prominent spatially organized patterns | Event coincidence analyses, SPI 1 month, soil moisture, atmospheric circulations and land-atmosphere feedbacks |
| De Luca et al., 2020 [43] | Concurrent wet and dry extremes | Median wet to dry transition is about 27 months and dry to wet is about 21 months | Land areas affected by extreme dry–wet anomalies are increasing with time | Global | PDSI, 1950–2014, dry–wet ratio, extreme transition time, teleconnections |
| Ford et al., 2021 [40] | Variability and rapid transitions of precipitation extremes | Large areas have had a significant increase in annual SPI range and associated magnitude of transition time | Findings are aligned with flash drought studies indicating a decrease in warning time | Midwest US | SPI, 30, 90 and 180-day scales, 1951–2019, thresholds for transitions |
| Chen Wang 2022 [18] | Transitions between dry and wet periods | Shorter dry-to-wet transitions are projected for 59% of the global land Variabilities accelerated the dry to wet transitions | Strong intensity and rapid transitions found in Eastern Canada and northern US | Global | SPEI 3-month observations 1954–2014, projections using CMIP6 |
| Rashid Wahl 2022 [44] | Consecutive dry and wet extremes | Consecutive dry-to-wet extremes are increasing over time | Numbers of events range from 20–30 in North America, 1901–2015 | Global | PDSI 6 months, multi-hazard risk, copula models, teleconnections |
| Rezvani et al., 2023 [21] | Projected compound droughts and floods (not precipitation) | Frequency of flood to drought events is projected to double at 1.5 °C global warming Transition time projected to increase | Climate warming increases variability, frequency and magnitudes Flood to drought transitions occur more quickly with climate change | Peace, Fraser and Columbia river basins in NW North America | Streamflow records and simulations, threshold method, Empirical Compound Severity Index, CMIP5 |

**Table 1.** *Cont.*

| Reference | Compound Extreme Type | Description /Finding | Temporal and Spatial Patterns | Region | Methods |
|---|---|---|---|---|---|
| Pokharel et al., 2023 [45] | Extreme spring dry-to-wet transitions One to two per decade in observations | Dry-to-wet transitions are likely to slow down and weaken in the future, especially after about 2050 | Dry–wet transitions are more common in the upper than lower basin in the observations | Colorado River Basin | PDSI, PHDI, SPI3 and 6 months, CMIP5 and 6 models, WRF PGW |

Note regarding acronyms in the table: SPI is the Standardized Precipitation Index, SPEI is the Standardized Precipitation Evapotranspiration Index, PDSI is the Palmer Drought Severity Index, PHDI is the Palmer Hydrological Drought Index, WRF PGW is Weather Research and Forecasting Pseudo Global Warming simulations, CMIP is the Climate Model Inter-comparison Project.

Researchers have examined a range of time sequences of CDP types, from pluvial ending droughts (e.g., [12,39,41]) to flash droughts ending pluvials (e.g., [19,22]) (Table 1). Several other characteristics are examined, including the number, duration and severity of these compound extremes.

The range of literature considers both past observations and reconstructions, as well as future projections. The latter tend to agree on the amplification of droughts and pluvials with drying regions having more pluvials [42], and greater variability with shorter and more intense dry-to-wet transitions [18]. Pokharel et al. found the opposite with decreased frequency and strength of dry-to-wet events projected with continued warming, especially after mid-century for the Colorado River Basin [45]. They provide ample reasons for these decreases, including the robust spring drying and shift of the North Pacific Subtropical High-pressure area. This is a reminder that different regions have different types of transitions in their climates and could likely have different climate futures.

Regarding past occurrences of CPDs, Christian found that the chance of a significant pluvial year after a significant drought year is about 25% in the Northern Great Plains of the US, adjacent to our study area [39]. Maxwell et al. used PDSI and other methods and found that the rapid termination of drought is more common than gradual endings [41]. He and Sheffield conclude that the transition from droughts to pluvials has increased in the past 30 years [12]. Ford et al. found an increase in the magnitude of transition time [40]. Rashid and Wahl determined that the number of transitions is increasing over time [44].

These studies used several water balance indices, such as the PDSI, SPI and SPEI, as well as some methods developed specifically for assessments of compound extremes. These methods include the wet/dry ratio [43] and the Event Coincidence Analysis [12].

This selected literature review gives a range of possible methods, comparisons and findings to further explore CDP and to design work in other regions, including Canada. It indicates increasing concern and interest in CDP, but also an emphasis on global analysis and a general lack of research, especially for the regional scale.

### 3.2. Analysis of the Risks of Past Compound Droughts and Pluvials in the Prairies

3.2.1. Convergence of Findings from Research Regarding Prairie Droughts and Pluvials

In this section, we compare and contrast the temporal and spatial characteristics of droughts and pluvials in the Prairies. First, we integrate the literature on Prairie droughts and pluvials to determine their nature as compound hazards. Then we combine the findings using the characteristics described in Table 2 for the period 1961 to 2022. This period was selected because of the availability of documentation of major and record pluvials and droughts in the Prairies. We describe the pluvials' characteristics, their relation to droughts and their stages, and the region(s) affected, as available. These examples were selected to demonstrate the relationships between the hydro-climatic extremes to characterize CDP.

**Table 2.** Combining literature sources to determine characteristics of some historical occurrences of CDP in the Prairies. Note that various stages of drought (onset, growth, persistence, and retreat) referenced here are taken from Bonsal et al. [46].

| Pluvial | Pluvial Characteristics | Relation to Drought | Region of Pluvial | References Combined |
|---|---|---|---|---|
| Record one-hour rainfall during drought | 250 mm in one hour in May 1961 | Record rainfall in the **growth** stage of a record drought of 1960–1962 | Southern Saskatchewan | Phillips 1993 [47]; Bonsal et al., 2011 [46] |
| Record eight-hour rainfall during drought | 375 mm in eight hours in July 2000 | Record rainfall in an **onset** stage (SPI) of record drought of 1999–2005 | Southwest Saskatchewan | Hunter et al., 2002 [48]; Bonsal et al., 2011 [46] |
| Major rainstorm during 1999–2004 drought | Intense rainfall, 8–11 June 2002 Two intense precipitation regions, one in the eastern and the other in the Western Prairies | Pluvial during the driest period of the drought, i.e., the **persistence** stage (PDSI), early **retreat** stage (SPI) and only three months after the peak of the drought | Southern Prairies from AB to MB, drought shifted northward | Szeto et al., 2011 [25]; Bonsal et al., 2011 [46] |
| 2010 | Spring and June 2010 Spring 2010 was the wettest in 63 years | Pluvials ended the meteorological drought of 2008–2010 with spring rains in 2010 | Northwest Saskatchewan in 2009, central and southwest Saskatchewan in 2010 | Wittrock et al., 2010 [49]; Hopkinson 2010 [50] |
| 2022 | Excess spring moisture | After a severe drought and heat dome in 2021 | Eastern Saskatchewan especially | AAFC 2023 [51] |

The Prairies region is home to record intense rainfall amounts. Several of these occurred during or closely associated with drought. For example, Canada's record intense one-hour rainfall of 250 mm occurred during May 1961 at Buffalo Gap in southern Saskatchewan [47] (Table 2). This was a compound extreme event because the year 1961 has been identified as the most extensive single-year Prairie drought of the twentieth century [52]. It also is the worst drought as measured by the severity and extensiveness of the August PDSI (12 month) for the 1900 to 2005 period [46]. The May 1961 extreme rainfall occurred during the growth stage of the drought according to the stages of drought as defined by Bonsal et al. [2,46] for a more precise exploration of the timing of the wet–dry connection. These stages are defined for a combination of severe drought or worse and for the percentage of grids within the agricultural region of the Canadian Prairie Provinces.

Another record rainfall was 375 mm in an eight-hour storm in July 2000 in the Vanguard area of southwest Saskatchewan [48] (Table 2). That storm is the largest area eight-hour event recorded in the Prairies. The amount of rain exceeded the average annual precipitation total of 360 mm for the area. Again, using the drought stage graphs of Bonsal et al. (2011) [46] and SPI only, July 2000 was classified as an onset stage of the record drought of 2001 to 2002. The PDSI indicated no stage for severe drought at that time, likely because of its lag effect.

Just a few years later during that same drought of 2001 to 2002, an intense and extensive rainstorm occurred in June 2002. That deluge brought 175 mm of rain to the Lethbridge area in Alberta and affected an area across the entire Prairies from western Alberta to Winnipeg, Manitoba [25]. That intense and extensive rainfall coincided with the early stage of retreat (using SPI) of the record 2001 to 2002 Prairie drought. However, using PDSI the rainfall coincided with the persistence stage of the drought, again using the stage information from Bonsal et al. (2011) [46]. The PDSI has a much greater lag effect and this would likely account for the difference in stages as the SPI would be more responsive to the pluvials.

The timing of these three examples of extreme rainstorms seems unusual as they occur during the onset, growth and persistence stages of major droughts. Many of the transitions assessed in the literature (Table 1) examined the dry-to-wet sequence, that is, wet periods at the end or terminating the drought. Szeto et al. suggested that the extreme drought of 2002 may have enhanced the probability of the June 2002 storm they documented [25]. We search for more examples of compound wet and dry events in the more recent period and characterize them for the Prairies in the next section.

3.2.2. Characterization of Compound Droughts and Pluvials in the Prairies Using the SPEI Global Drought Monitor, 2000 to 2022

Next, we address questions about the characteristics of the CDP and selected events in the Prairies using the SPEI Global Drought Monitor, which is a suitable source for this initial exploration [22,30,53]. These questions include the locations, intensities and spatial patterns of drought and pluvials, and indications of the strength of the gradients and the orientation of the boundary zone between wet and dry conditions (Table 3). General locations are provided in Table 3, but the SPEI Monitor can provide much more specific locations because of its one-degree by one-degree spatial resolution and accessible grid values.

**Table 3.** Selected examples of the relationship of main drought and pluvial conditions during 2000 to 2022 in the Prairies. using the SPEI Global Monitor for 12-month August value (i.e., 12 months ending in August) to represent the agricultural year (http://spei.csis.es/, accessed various dates July to September 2023 [30]).

| Dates | Main Areas of Drought | Main Areas of Wet Conditions | Relationships |
|---|---|---|---|
| 2000 | Drought in S AB, most intense at SPEI −2.0 in central AB | Wet with SPEI +1.4 in SE MB | A strong west to east gradient from severe drought in AB to normal in central SK. Boundary between dry-to-wet is oriented N to S and located west of the AB-SK border |
| 2001 | Drought has intensified in AB and is worst in central N SK at SPEI −2.3 | Wetter in much of MB to +1.6 in central MB | The west to east dry-to-wet gradient has intensified and shifted westward. The dry–wet boundary is oriented N to S, and is near the SK-MB border |
| 2002 | Drought is most intense at −2.0 in W AB and drought extends across central SK and into MB | Wet in S AB and central and S SK at +1.6 | Patterns have switched as drought migrated northward. Pattern is dry in central areas of AB and SK and very wet in the south Dry-to-wet boundary is oriented W to E |
| 2009 | Intense drought in central and northern AB at −1.7, and has ended mostly in SK and MB | Near normal conditions in SW AB across to MB with wetter in E MB to +2.2 | West to east dry-to-wet gradient is at the AB-SK border in central areas and into SK Dry-to-wet boundary is more complicated. It is oriented N to S in central areas, and is in western SK Another boundary exists from central to southern SK |
| 2010 | Drought has receded with near normal to drought of −1.5 in W AB | Normal to wet at +2.0 in W AB, wet most of SK to more than +2.3, normal to wet in N MB at more than +2.0 | Severe to extreme wet over much of the Prairies, except western AB Strong dry-to-wet boundary is oriented mostly N to S in central AB |
| 2011 | Some pockets of dry to drought exist in central MB. No large areas of drought | Very wet in S SK at 2.3 in SE, normal to wet in AB | Boundary of dry to normal is along the SK MB border in a mostly N to S alignment |

**Table 3.** *Cont.*

| Dates | Main Areas of Drought | Main Areas of Wet Conditions | Relationships |
|---|---|---|---|
| 2015 | Extreme drought in AB with worst at −2.2, extending across SK with worst at −1.7 and into MB at −1.4 | NE SK at +1.9 and NW MB are the only wetter regions | Extensive drought across the Prairies Strong dry-to-wet boundary in NE SK with NW to SE orientation |
| 2022 | Drought mostly concentrated in AB and SK at −1.5, with near normal in central and NW AB, near normal to wet in E MB | Wet at +2.2 in E MB | SK drought is sandwiched between near normal in W AB to wet in MB Strongest dry-to-wet boundary is in W MB and is oriented N to S |

Notes for the table: We apply the classifications used for SPI in Bonsal et al. (2011) [46] for SPEI. That is, near normal is SPEI > −0.5 to 0.5, severe drought is >−2.0 to −1.5, and severe wet is +1.5 to <+2.0. We use the abbreviations for the Prairie Provinces, AB is Alberta, SK is Saskatchewan, and MB is Manitoba. We use abbreviations for the directions, S is south, N is north, E is east, and W is west, and so on.

These examples of pluvials in conjunction with droughts possibly have the more common timing occurring at the end of a drought and playing important roles in alleviating or even terminating the droughts. Unfortunately, drought stages have not been determined for these events. Wheaton et al. provided an overview of the documentation of past extreme precipitation events in Saskatchewan [54]. For the study period, they noted several examples of wet years, including 2000, 2002, 2010 to 2012. They also noted that torrential rainstorms have occurred during major droughts and that severe droughts have shifted to very wet conditions within days.

An understanding of the temporal relationships of these dry–wet and wet–dry dynamics is a scientific basis for risk assessment. Selected examples of the characteristics of the DP interactions during 2000 to 2022 are summarized in Table 3. Dates were selected based on our knowledge of the major droughts and pluvials in the Prairies and their stages (e.g., [2,46,54]). The main areas of drought tend to be in Alberta and Saskatchewan, although the 2002, 2015 and 2021 droughts extended across the Prairies from Alberta into Manitoba. The 2021 drought is not a CDP as no pluvials existed with this SPEI value. The 2002 drought is an unusual case as the strong pluvial in the southern Prairies was associated with a northern migration of the drought area.

The locations of pluvials appear to be more variable. However, the most common pattern is associated with wetter areas in MB, and dry in the west to wetter in the east. This orientation of the dry-to-wet gradient reflects the gradients in precipitation normals. Boundaries between the drought and pluvials show variation, but these examples mostly have a north-to-south orientation. These patterns are an indication of how the CDP can change with intensities and locations, as well as the rate of change over space.

The SPEI12 month for 2002 is a case of the more unusual example of the dry-to-wet boundary zone that is oriented west to east (Figure 2). It is also a good example of a strong gradient, that is rate of change, from wet in the south to dry farther north. It also shows the spatial pattern of the CDP resulting from the major rainstorm in the southern prairies in the summer of 2002 and the shifting of the drought northward. These patterns are a contrast to the more usual patterns described in Tables 2 and 3.

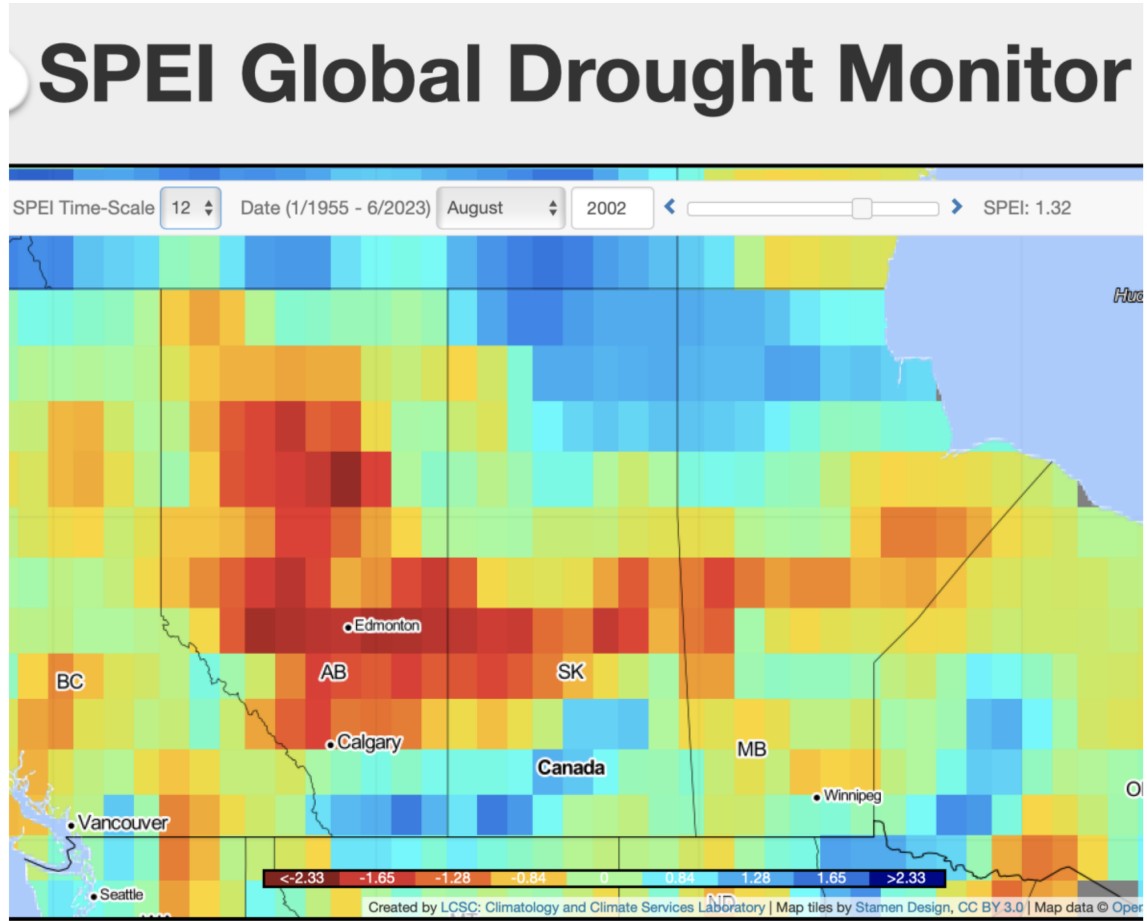

**Figure 2.** Patterns of the 2002 compound drought-pluvial event in the Canadian Prairies using the SPEI Global Monitor, SPEI12month for August (https://spei.csis.es/, accessed on 5 August 2023 [30]). Note that this map is a screen capture and that the quality of the map has been retained to illustrate the map type used.

## 4. Discussion and Conclusions

The purposes of this paper were (1) to compare and synthesize recent literature concerning the past and future possible risks of compound droughts and pluvials with relevance to the Canadian Prairies, and (2) to explore and characterize the compound risk of drought and pluvial events in the Prairies using various data and sources. Examples are used to further advance knowledge of the risks and to enhance adaptation to these compound extremes.

The first purpose addressed the question, what literature has examined the compound characteristics of both droughts and pluvials? Our review of the literature regarding CDP found that research examining these extremes as separate hazards, is much more common and more advanced than studies of their joint occurrence which is quite limited. Only a few papers assessed these compound extremes on the Prairies and over only limited time periods. With continued climate change, the threat of compound droughts and pluvials is an increasing concern.

Many more examples of research of other types of compound extremes were found for other regions, especially drought and heat events. A few papers dealt with CDP in the United States, and more papers were for the global scale. Ten examples of relevant literature were described according to the type of compound extreme, findings, spatiotemporal scales, regions and methods: in the context of the Great Plains of the United States, findings such as increases in the frequency of dry-to-wet events [39], and rapid transitions of wet-to-dry [40],



and dry-to-wet [12]. In the Canadian context, only four papers were found that examined some forms of CDP. They are limited to two main drought events and to the Prairies.

The second purpose was to address the question, what are some examples of CDP in the Prairies? We compared and contrasted findings from integrating literature that considered Prairie drought and pluvials separately and from examining patterns of the Global SPEI Monitor to focus our analysis on the more recent past (2000 to 2022). Several record to near-record rainfalls were found to be associated with severe droughts. For two of the examples the pluvials occurred during the early stages of the droughts, while one occurred during the persistence stage and the others occurred at the end of the droughts. These examples demonstrate the characteristics of the extreme pluvials, their relationship to severe droughts, and the regions affected. Further work could be carried out to find more examples in order to determine further characteristics of CDP. Also, developments and applications of quantification of CDP could be made.

Several more examples of CDP were found using some main severe droughts as a starting point and using the Global SPEI Monitor. Main areas of wet conditions and their spatio-temporal relationships with the droughts were documented. Although the core areas of drought for the selected examples tended to be in the western portion of the Prairies, the locations of the associated pluvials appeared to be more variable. The most common pattern of the CDP events is wetter in the east to drier in the west. The boundaries between the drought and pluvials tend to be oriented north to south.

We have demonstrated through the lens of CPD that variability dominates the hydroclimate of the Canadian Prairies. The variability of droughts and pluvials is likely to increase in many regions globally with climate change, according to most of the literature reviewed. Climate change is a critical driver of the changing characteristics of both drought and pluvials, therefore understanding of this effect is essential.

We emphasize the need to more fully understand the characteristics and risks of the compound extremes of droughts and pluvials, as well as important ways to decrease vulnerability and associated damages. Therefore, improved methods to characterize and to quantify CDP are required. This paper is the first to explore the concept of and many examples of CDP for Prairies and for Canada. Although the study area is the Canadian Prairies, the work is relevant to other regions that are becoming more vulnerable to increasing risks of and vulnerabilities to such compound extremes.

Enhanced monitoring of both droughts and pluvials is critical. The Canadian and North American Drought Monitors are very useful, but should be expanded to include pluvials such that the compound nature of these extremes is better documented and understood. This need is becoming even more pronounced with continued global warming. Understanding of drought is considerably advanced by considering precipitation patterns (or lack of those) that shape the drought's beginning, ending and other stages. The reverse holds true for wet periods as those also are shaped by interaction with dry conditions.

Gaps of knowledge abound and include the assessments of the characteristics, impacts, adaptations, and vulnerabilities associated with CDP. The recent severe and large area drought of Western North America in 2021 is a prime example. With the ending of projects such as the Drought Research Initiative [38], new projects focused on droughts and pluvials are needed to advance understanding. Research on future characteristics of these extremes tends to emphasize the ensemble or median results. Worst-case possibilities also should be examined for appropriate risk assessment.

**Author Contributions:** Conceptualization, E.W., B.B. and D.S.; Methodology: E.W. and B.B.; Data curation and analyses: E.W.; Writing, original draft preparation: E.W.; Writing-review and editing: E.W., B.B. and D.S. All authors have read and agreed to the published version of the manuscript.

**Funding:** This research received no external funding.

**Data Availability Statement:** Data links supporting reported results are included in the manuscript.

**Acknowledgments:** The authors thank Darrell Corkal for discussions. We thank the anonymous reviewers for their reviews and constructive comments that helped improve the manuscript. We are thankful for the use of the maps from the Global SPEI Drought Monitor.

**Conflicts of Interest:** The authors declare no conflict of interest.

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
