# Peer review of "Compound Extremes of Droughts and Pluvials: A Review and Exploration of Spatio-Temporal Characteristics and Associated Risks in the Canadian Prairies"

_water, doi:10.3390/w15193509_

Round 1

Reviewer 1 Report

I received the paper titled Compound extremes of droughts and pluvials: a review and exploration of spatio-temporal characteristics and associated risks in the Canadian Prairies to make the report.

After reading the text, I formulated some suggestions and advice for the authors.

In the introduction, I propose to place a map of Canada with the place that is being discussed (prairies) marked - Alberta, Saskatchewan and Manitoba, termed Prairies here. The Figure 1 at the end of the article is hard to read (low quality).

Authors mentioned about different types of droughts (line 46-49). I suggest to indicate, even in brackets the types of droughts. Please check the table 8 in the paper: Water and Hydropower—Challenges for the economy and enterprises in times of climate change in Africa and Europe. Water14(22), 3631.

Line 109 – Line 109 - The authors have included research questions. Please answer these questions in the discussion section.

Line 123 – Why did the Authors not use the Global SPEI Monitor primary source but a secondary source? I suggest to use https://spei.csic.es/

In the methodological part, I propose to write what methods were used. In the science these methods are described. I suggest, as before, to read the article referred to above (Water and Hydropower—Challenges for the economy and enterprises in times of climate change …) and use the information about the methods used. Authors can use also  What are different research approaches? Comprehensive Review of Qualitative, quantitative, and mixed method research, their applications, types, and limitations. Journal of Management Science & Engineering Research5(1), 53-63.

What is the scale of numbers in Figure 1? Range from -2.33 to +2.33. In addition, I suggest a better quality of this Figure. For better quality, use a paintbrush or other program.

In your next version of the paper, please use the mdpi style of citations

After that I hope the paper will much better. Good luck

Author Response

A map of Canada with the Prairie Provinces and agricultural area highlighted is included.

The different types of droughts are now indicated along with references, including the suggested reference.

The research questions beginning line 109 are now addressed more clearly in the Discussion and Conclusions Section.

Line 123 The Global SPEI Monitor primary sources was used at https://spei.csis.es. We now have specified this site in the figure caption. We have also specified SPEI references stated in the SPEI section 3.2.2, namely Vicente-Serrano et al. (2010) and Begueria et al. (2010). These references are referred to in the first paragraph of section 3.2.2.

The methodology section is expanded and the methods are described more fully using the suggested articles.

The scale of the SPEI numbers of Figure 1 (now Figure 2) is explained in the methods with examples. This explanation augments the notes for Table 3 that outlined the classifications for SPEI.

 The SPEI map (Figure 2) is a screen capture. We have decided to retain the quality of the map to illustrate the type of maps used.

The MDPI style of citations with numbering in the text and the reference section is used. This is shown in the version of the paper labelled references done.

Thanks for the review.

Reviewer 2 Report

In general a well-written manuscript with only one minor drawback. The Discussion and Conclusions section is a bit sparse and would benefit greatly  by expansion to place the available information in greater context for Canada and other prairie ecosystems. 

A lot of "gray" literature. Might help if more open literature is available but this may be it.

Minor edit: 

Line 372: Replace "that examined" with , examining"

Line 373: Place comma after hazards

Author Response

Replies to Reviewer Two

The Discussion and Conclusion Section has been expanded. It now places the information in greater context for Canada and for the prairie ecosystems in the United States.

Yes, a lot of gray literature has been used, as needed, but many scientific journal articles have also been used. The gray literature is often available from the respective government sites.

The minor edits have been made.

The MDPI style of citations with numbering in the text and the reference section is used. This is shown in the version of the paper labelled "references done".

Thanks for the review.

Reviewer 3 Report

The idea of the paper is good. However, appropriate revisions to the following points should be undertaken in order to justify.

Major revisions are needed.

The abstract needs more enhancement. Please re-write an abstract section, explain an obtained result and contribution, improve a proposed method, etc.

1. The advantages and limitations of the proposed approach in relationship with similar schemes is not clear.

2. Please revise the structure of the paper.  Suggest the discussion as separate Section.

3. The paper is poorly written and should be substantially revised. For instance, in line 143 missing punctuation comma after Szeto at al. (2011); in line 261, SPI), “)” is redundant.

4. The image in Figure 1 is not clear. Please provide a higher resolution image.

5. In the methodology section, the authors are encouraged to provide more comprehensive details about each step of the CDP.

6. Please carefully check and revise the references list according to the Instructions for Authors of this journal.

7. Authors should add more references in literature such as https://doi.org/10.1016/j.wse.2018.10.003

The language used in the paper needs improvement. 

Author Response

The abstract has been changed to better explain the results, contribution and to refer to improved methods.

  1. The advantages and limitations of the approaches are more clearly stated.
  2. The Discussion and Conclusions Section has been expanded. We kept the section as one part for streamlining and to avoid duplication. This is a common structure in many journal articles.
  3. The punctuation problems were fixed and the entire manuscript was checked and improved regarding further punctuation and language issues.
  4. The SPEI map (now Figure 2) is a screen capture. We have decided to retain the quality of the map to illustrate the type of maps used in the methods of the paper.
  5. More comprehensive details of the steps of the compound drought and pluvial methods are provided in the methodology. More references have been added.
  6. The MDPI style of citations with numbering in the text and the reference section is used. This is shown in the version of the paper labelled references done.

  1. The suggested reference has been used in the methodology section. The reference supplements the description of the methods.

Re improvement of the quality of the English Language: The authors have corrected the punctuation, as requested, and we have also improved other issues of grammar and punctuation.

Thanks for the review.

Round 2

Reviewer 1 Report

Dear Authors,

Thank you for your revised paper. Now it is much better. It can be published.

I suggest to read some scientific papers, where different types of methods were described in the systematic way (e.g. Significance and Directions of Energy Development in African Countries "Energies"). It will help to organize future resaech.

Reviewer 3 Report

The revised paper has addressed all my previous comments.I believe the manuscript has been sufficiently improved to warrant publication in Water.